# Have Diagnostics, Therapies, and Vaccines Made the Difference in the Pandemic Evolution of COVID-19 in Comparison with “Spanish Flu”?

**DOI:** 10.3390/pathogens12070868

**Published:** 2023-06-23

**Authors:** Florigio Lista, Mario Stefano Peragallo, Roberto Biselli, Riccardo De Santis, Sabrina Mariotti, Roberto Nisini, Raffaele D’Amelio

**Affiliations:** 1Istituto di Scienze Biomediche della Difesa, Ispettorato Generale della Sanità Militare, Stato Maggiore della Difesa, 00184 Roma, Italy; 2Centro Studi e Ricerche di Sanità e Veterinaria, Comando Logistico dell’Esercito, 00184 Roma, Italy; 3Ispettorato Generale della Sanità Militare, Stato Maggiore della Difesa, 00184 Roma, Italy; 4Dipartimento di Sanità Pubblica e Malattie Infettive, Sapienza, Università di Roma, 00161 Roma, Italy; 5Dipartimento di Malattie Infettive, Istituto Superiore di Sanità, 00161 Roma, Italy; 6Dipartimento di Medicina Clinica e Molecolare, Sapienza, Università di Roma, 00198 Roma, Italy

**Keywords:** Spanish flu, COVID-19, influenza virus, coronavirus, SARS-CoV-2, pandemic, vaccines

## Abstract

In 1918 many countries, but not Spain, were fighting World War I. Spanish press could report about the diffusion and severity of a new infection without censorship for the first-time, so that this pandemic is commonly defined as “Spanish flu”, even though Spain was not its place of origin. “Spanish flu” was one of the deadliest pandemics in history and has been frequently compared with the coronavirus disease (COVID)-19 pandemic. These pandemics share similarities, being both caused by highly variable and transmissible respiratory RNA viruses, and diversity, represented by diagnostics, therapies, and especially vaccines, which were made rapidly available for COVID-19, but not for “Spanish flu”. Most comparison studies have been carried out in the first period of COVID-19, when these resources were either not yet available or their use had not long started. Conversely, we wanted to analyze the role that the advanced diagnostics, anti-viral agents, including monoclonal antibodies, and innovative COVID-19 vaccines, may have had in the pandemic containment. Early diagnosis, therapies, and anti-COVID-19 vaccines have markedly reduced the pandemic severity and mortality, thus preventing the collapse of the public health services. However, their influence on the reduction of infections and re-infections, thus on the transition from pandemic to endemic condition, appears to be of minor relevance. The high viral variability of influenza and coronavirus may probably be contained by the development of universal vaccines, which are not easy to be obtained. The only effective weapon still remains the disease prevention, to be achieved with the reduction of promiscuity between the animal reservoirs of these zoonotic diseases and humans.

## 1. Introduction

Since the first description of the new coronavirus severe pneumonia in December 2019 and the subsequent declaration of pandemic by the World Health Organization (WHO) on 11 March 2020, different papers compared the 1918 “Spanish flu” and the 2019 COronaVIrus Disease (COVID-19) pandemics, in order to infer a possible prediction of any epidemiological and clinical trends of COVID-19 [1,2,3,4,5,6,7,8,9]. These studies have been developed in the early phase of the new pandemic, before the description of new characteristics and complications of COVID-19. This study aims, therefore, to deepen this comparison and speculate on the role that innovations after one century of scientific and medical progress might have had in the pandemic evolution.

## 2. The Viruses

Two different single-stranded (ss)-RNA viruses caused the “Spanish flu” and COVID-19 pandemics: influenza virus, that pertain to the family *Orthomyxoviridae* [10], and Severe Acute Respiratory Syndrome CoronaVirus (SARS-CoV)-2, classified in the family of *Coronaviridae* [9], respectively.

Four types of influenza virus are described, A, that may be responsible for epidemics and pandemics, B and C, which may be pathogen for humans, and D [11]. Coronaviruses include α, ß, γ, and δ genera, but only α, ß may infect humans.

Influenza is a negative-sense, single stranded (ss)-RNA virus and its genome is segmented (eight different gene segments in types A and B, seven in types C and D). Two of these segments in the type A code for either 18 different hemagglutinins (HA or H) and 11 different neuraminidases (NA or N); the two glycoproteins needed for attachment to and detachment from the host cells, respectively [11,12]. The natural reservoir for influenza virus is represented by the waterfowl birds (excepting HA17NA10 and HA18NA11, which have only been observed in Peruvian bats); however, even mammals, including humans and swine, may be infected [13]. Among the theoretical 198 possible subtypes, 120 have been found in nature, and only three, namely H1N1, H2N2 and H3N2, have been observed in humans in three different pandemics, thus demonstrating a stable adaptation to humans [12]. The virus enters the host cell via a cellular surface receptor of sialic acid (Sia), with a link to galactose, which is α(2,3) in avian-type and α(2,6) in human-type. In humans, the Sia α(2,6) is located in the upper airways, whereas Sia α(2,3) is present in the lower airways; a possible adaptation of an avian influenza A virus to humans, therefore, may maintain its capacity to infect the cells of the lower airways, inducing a very severe pneumonia with possible acute respiratory distress and high case-fatality rate.

Conversely, coronaviruses are positive-sense ss-RNA viruses with a compact, unsegmented genome, which enter the host cells mostly via the angiotensin-converting enzyme (ACE)2 receptor that is bound by the Spike protein (SP), which is a characteristic molecule of coronaviruses, through its receptor binding domain (RBD) located in the S1 tract. ACE2 receptors are expressed on cells of many organs, including the respiratory tract, at the level of both higher and lower airways [14]. Recently, it has been suggested that coronaviruses may also bind Sia receptors, similarly to influenza viruses [15]. COVID-19 is a zoonosis, but the animal reservoir has not yet been precisely identified, even if the major suspects are bats [16]. Coronaviruses may be responsible for the common cold, such as the α-coronaviruses 299E and NL63, and the ß-coronaviruses HKU1, OC43, but may be involved in severe pneumonias and high mortality rate, as in the case of the ß-coronaviruses Middle East Respiratory Syndrome coronavirus (MERS-CoV), SARS-CoV and SARS-CoV-2, with only the last one that was able to induce a pandemic [17].

Although structurally and genetically different, SARS-CoV-2 and influenza virus share similar characteristics. Both viruses show high a capability to generate variants, which may be antigenically and biologically different from the viral strain originally appeared, as in the case of SARS-CoV-2, or from the viral strains circulating in previous years, as in the case of influenza virus. This characteristic makes them able to infect a higher number of individuals, and potentially cause pandemics, in comparison to viruses with a more stable genome. In fact, antigenic variability makes the specific immunologic memory less or not at all efficient, also rendering susceptible to infection those individuals that had previously been infected with the same virus. The rate of variability is quite high for both viruses, as in general for RNA viruses, whose RNA polymerase has a very low capacity to check for possible replication mistakes than the DNA polymerase, which is present in DNA viruses, that are, therefore, more stable. The genome mutation rate for human influenza A virus has been calculated as 5.7 × 10^−3^ substitutions/site/year [18], whereas in SARS-CoV-2 it has been calculated as 1.12 × 10^−3^ mutations/site/year [19]. Although the RNA polymerase of all coronaviruses, including SARS-CoV-2, has proofreading exonuclease activity [20], this does not seem to markedly reduce the viral variability, considering that the genome mutation rate appears similar to that of influenza A virus. Variants generated by the influenza virus are principally a consequence of intracellular viral genetic reassortment, whereas variants of coronaviruses are nearly exclusively a consequence of recombination, with the additional possibility of point mutations in both [21]. The “antigenic shift” or reassortment variations in influenza A, may be more or less marked, involving all or some gene segments. It is a deep viral genetic modification associated with pandemics occurring with a variable periodicity. Recombination and reassortment are generated by genetic material exchange in cells infected by different viruses. In the subtype H1N1 “Spanish” pandemic, all gene segments were completely new, whereas in the subtype H2N2 Asian pandemic in 1957 and in the subtype H3N2 Hong Kong pandemic in 1968, only three and two new gene segments were involved, respectively [22]. Finally, in 2009, the A(H1N1)pdm09 flu pandemic was caused by a swine virus coming from the reassortment of six gene segments from a triple reassortant swine virus (in which five gene segments of a North American swine origin virus reassorted with gene segments from avian and human origin) and two gene segments coming from the A(H1N1) Eurasian swine virus lineage [23]. This is a clear example of the complexity of intra- and inter-species reassortment of influenza A virus. The emergence of the new subtypes in the pandemics is generally accompanied by the disappearance of the previous subtype [23]. H1N1 disappeared from humans in 1957, when it was replaced by the H2N2 Asian pandemic, which was in turn replaced by H3N2 in 1968 during the Hong Kong pandemic. However, H1N1 suddenly re-emerged in 1977, probably in consequence of accidental release from a laboratory source, causing the Russian flu pandemic in 1977, and since then is co-circulating with H3N2 [23]. The “antigenic drifts” are variations caused by limited mutations of antigenic determinants that occur frequently, almost yearly, and are associated to seasonal epidemics, but generally not to pandemics, since the viral antigens are not completely changed, and the immunologic memory makes a large part of the population less susceptible to infection. The current circulating influenza A virus subtypes in humans are H1N1 and H3N2, but some avian flu viruses, such as A(H5N1), A(H7N7), A(H7N9), and A(H9N2) have been observed in limited outbreaks, thus allowing to infer that no adaptation to humans has still been achieved by them [https://www.who.int/news-room/fact-sheets/detail/influenza-(avian-and-other-zoonotic), accessed on 10 April 2023].

Coronaviruses present a quite frequent recombination rate [21] and, in fact, many variations in SARS-CoV-2 genome were observed and the most potentially dangerous are defined as variants of concern (VOCs). The VOCs observed up to now are Alpha (B.1.1.7), Beta (B.1.351), Gamma (P1), Delta (B.1.617.2), and Omicron (B.1.1.529), which is still largely present with several subvariants [24], whereas the previous VOCs have virtually disappeared (Table 1).

The zoonotic origin of the viruses responsible for the two pandemics is unknown, but the genomic variability is certainly involved in their biologic success as pathogens. For Spanish flu, an avian strain of influenza virus adapted to mammals, human or swine, may only be hypothesized, since no data exist on previous circulation of the viral strain, making any attempt to reconstruct the viral origin impossible [25]. Various hypotheses on the origin of SARS-CoV-2 have been proposed, but no definitive answer is still available [16,26].

The capacity of the two viruses to cause severe diseases is similar, but with different immunopathogenic mechanisms. In fact, in the large majority of patients with “Spanish flu”, the viral infection favored the development of a bacterial disease that nearly constantly ended with death in that pre-antibiotic era, while an isolated viral pneumonia was more rarely observed [27]. In COVID-19, instead, a virus-induced hyper-activation of the immune system, with consequent cytokine storm and immunopathogenic effect at lung level and in the whole body, with the picture of multi-organ failure, is generally found in severe cases [7].

## 3. Epidemiology

“Spanish flu” occurred in three epidemic waves, the first in the spring/summer 1918, the second in the fall 1918, and the third in the winter 1918/1919. The first wave widely spread throughout the United States, Europe, and the rest of the world: its morbidity rates were generally high, but the death rate was estimated to be at 0.1% [28], not different from the usual death rates observed among influenza patients in previous years. The second wave burst between September and November 1918, with a death rate significantly higher than that observed in the first wave and estimated at 2–4% [28], because of the higher frequency of complicated, severe, and fatal cases. A third wave occurred in the first months of 1919 in many countries with different patterns: this wave had high illness rates in part of European countries, such as France, Scotland and Finland, lower rates in Sweden, Norway and Holland, while it was almost absent in Spain, Denmark and Italy. Generally, the third wave was less intense in those geographical areas where the second wave was more severe [29], and its mean death rate was 1% [28]. It was hypothesized that a different virus could be responsible for the second wave, but the observation that those who survived the infection in the first wave resulted protected during the second wave seems to limit the probability of this hypothesis [8]. However, it has recently been suggested that a second antigenically undistinguishable virus might have circulated in the second pandemic wave, in addition to the one in the pre-pandemic period [28], that could have evoked cross-protective antibodies. In line with this hypothesis, it has recently been proven that a single amino acid substitution in the sequence of H1 may increase its affinity for the Sia α(2,6) receptor [30], and increase the adaption of a flu avian viral strain to humans. In the analysis of a series of stored lung tissues from US military personnel who died for influenza in the pre-pandemic and pandemic (second wave) 1918 periods, a trend from an “avian-like” to a “human-like” pattern of H1-Sia receptor was observed, in the pre-pandemic and pandemic period, respectively [31]. The “Spanish” definition of this pandemic does not reflect the real geographic origin of the pandemic, which seems rather to be born in a military training camp in the USA, or Europe (France or England) [32]. Although reliable data are not available, it is estimated that “Spanish flu” infected approximately 500 million subjects, one third of the world’s population in 1918, with a mortality of approximately 50 million cases [33]. According to other estimates, real pandemic mortality might have fallen between 50 and 100 million deaths [34]. The reliability of these estimations is limited considering the ignorance of the etiological viral agent, thus the impossibility of confirming diagnosis, and collecting reliable epidemiological data, in a period of general social disruption in consequence of the war. In addition to the war conditions, characterized by overcrowding, poor sanitation and straining of the health services, all favoring the spreading and severity of the infection, even scientific accuracy of data collection could be limited by military and political considerations. Finally, other war-independent variables could influence mortality, with an inverse association between mortality and socio-economic status [35], and a high variability of case-fatality rates according to geographical places, age, sex, and ethnic group [36]. All these considerations also make the estimates of global case-fatality rates less reliable [33]. It is estimated that in 1918, 56,000 American soldiers died for “Spanish flu” in Europe and in the training camps in the USA, versus 53,402 killed in combat [37]. One peculiar characteristic of “Spanish flu” was the epidemiological curve of deaths, being W- instead of U-shaped, as usually observed in the seasonal epidemics and in other flu pandemics. The fact that older adults fared better than younger adults in influenza severity might be due to the presence among older adults of an immunological memory to an H1 influenza A virus that was circulating in 1889, thus causing a lower case-fatality rate among subjects aged 30–60 years than among those aged 18–30 years [38]. In fact, generally the extreme ages of life are particularly burdened by mortality, whereas the middle, productive, age of young adults is spared. Conversely, in the H1N1 “Spanish flu” pandemic, the middle-aged people were not spared and the group of persons < 65-year-old had >99% of flu-related deaths, versus 36% in the H2N2 Asian pandemic of 1957 and 48% in the H3N2 Hong Kong pandemic of 1968 [33].

After more than one century from “Spanish flu”, the epidemiological situation of COVID-19 has carefully and daily been monitored, by the Johns Hopkins University, Coronavirus Resource Center, up to 10 March 2023, three years from the pandemic declaration; up to this date, the total infected people were 676,609,955 and the total deaths 6,881,955, with a global case-fatality rate of 1.01% (https://coronavirus.jhu.edu/map.html, accessed on 10 March 2023). Moreover, at the same date and according to the same data source, 13,338,833,193 vaccine doses have been administered to the world population and the number of waves of disease have been six, half of which occurred in 2020–2021 and the other three in 2022, in coincidence with the spreading of Omicron variant. Similar figures have also been monitored by the WHO, which declared end to the pandemic as a public health emergency on 5 May 2023. As of 3 May 2023, 765,222,932 cumulative cases and 6,921,614 deaths worldwide were registered; as of 30 April 2023, 13,344,670,055 vaccine doses have been administered, 5,106,051,703 persons fully vaccinated and 5,548,001,227 persons vaccinated with only one dose have been calculated (https://news.un.org/en/story/2023/05/1136367, accessed on 3 June 2023). The highest wave of disease was the first one of 2022, whereas the highest mortality has been observed in the second half of 2020 and in the first half of 2021, before full implementation of the vaccine campaign. These data are in line with what has been observed in several studies, showing that COVID-19 vaccines are scarcely effective in preventing infections and in interrupting virus circulation, while they are very effective in reducing mortality. However, there are many other factors, including naturally acquired immunity and the spreading of milder viral variants, that may contribute to reduce mortality. Unlike “Spanish flu”, COVID-19 generally spares children and young adults, and shows its highest mortality in the >65-year-old people, especially if they are carriers of comorbidities [7].

## 4. Public Health Containment Measures and Therapeutical Approaches

In both pandemics, although they were spaced apart of over one century, the hygiene measures to try to limit the spreading of the infection were the same: social distancing, protective masks for respiration, isolation and quarantine in case of symptomatic infection. The possible transmissibility of SARS-CoV-2 before the onset of symptoms [39] may have reduced the effectiveness of these measures in COVID-19. However, the rapid availability of accurate and quick molecular diagnostic tools may have precisely driven the adoption of the public health measures reported above, thus increasing their effectiveness.

Empirical therapeutical approaches were adopted in both pandemics, including the administration of medicines with unproven efficacy, such as quinine during “Spanish flu” and hydroxychloroquine during the first phase of COVID-19 [40]. Another analogy is the use of passive immunotherapy, which was discovered by Emil von Behring and Shibasaburo Kitasato in 1890 as an anti-toxic treatment of tetanus and diphtheria; during the World War I hyper-immune heterologous serum prophylactic administration resulted to be significantly protective against tetanus [41]. In “Spanish flu”, the observation that the survivors of the first wave were protected during the severe second wave suggested the feasibility of immunotherapy through the plasma of convalescent patients, which was successfully used, then as the only possible specific therapy [42,43,44,45,46,47,48]. A similar approach was adopted in COVID-19 with plasma from convalescents, and showed a certain efficacy, particularly if administered in the first week from the start of symptomatology, and represented a precious therapeutical resource to reduce disease severity and mortality in the first phase of the pandemic, when other preventive or therapeutic tools were not yet available [49,50,51,52,53,54,55,56,57].

Conversely, the most significant difference between the two pandemics was the much more advanced technology and scientific knowledge in the year 2020 than in 1918. In addition to the availability of antibiotics and life support techniques in intensive care units of hospitals, the research and the industries made accessible rapid and precise diagnostic tools shortly after the identification of the pathogen during COVID-19 pandemic. The etiological diagnosis paved the way to develop, in rapid succession, antivirals and monoclonal antibodies, as well as different types of specific and effective anti-viral vaccines, which were made available at a large scale for global immunization in less than one year.

## 5. Diagnostic Tools, Antibiotics, Antivirals, Monoclonal Antibodies, Vaccines

### 5.1. Diagnostic Tools

Following the official notification of the SARS-CoV-2 genomic sequence, in most countries the molecular diagnostic tests for viral identification on nasopharyngeal and oropharyngeal were promptly set up. The Real Time Reverse Transcriptase Polymerase Chain Reaction (RT-PCR) is the most frequently used molecular technique for diagnosis of SARS-CoV-2 and represents the diagnostic golden standard. Primers and probes were designed to detect specific targets or regions of interest (ROIs), the most used of which are *ORF1ab/RdRp*, *E*, *N*, and *S genes* [58,59,60,61].

The availability of recombinant SARS-CoV-2 antigens made it possible to produce serological tests, based on the identification of specific antibodies, that could be used after at least one week from the onset of symptoms. The SARS-CoV-2 antigens included in the serologic diagnostic kits are S or nucleocapsid (N) proteins [59]. More recently, after the diffusion of vaccination, the use of these two antigens allow to discriminate vaccine-induced (positive for anti-SP only) from infection-induced antibodies (positive for anti-N and/or anti-S protein). Serological tests may help to identify a previous inapparent contact with SARS-CoV-2, when the virus is cleared, the intensity of the antibody response to the vaccine and may be useful for epidemiology studies of prevalence in defined populations.

With the production of SARS-CoV-2 specific antibodies, more rapid, economic, and manageable antigenic tests, some of which even self-administered, were set up and commercialized [58,59]. These tests are approximately one thousand-fold less sensitive than the molecular ones (the sensitivity is 10^5^–10^6^ viral copies per ml versus 10^2^–10^3^/mL, respectively); however, such a different sensitivity does not appear to represent a practical problem, considering that patients with <10^6^ viral copies per ml may hardly transmit the virus [59].

The availability of these tests makes the deepest difference in the efficacy of pandemic preventive measures between COVID-19 and the “Spanish flu”, whose diagnosis was only based on the clinical history analysis and the clinical observation, with the support of the lung X-ray, which debuted in the World War I [41]. No laboratory support to confirm the diagnosis of the virus was available, since the influenza virus was unknown in 1918 and would have only been identified 15 years later, in 1933 [62]. Accordingly, the public health measures, which are almost the same at one century of distance, are more effective in circumscribing the infective focus or cluster, if driven by rapid and precise identification of infected patients and in contact tracing, particularly important in COVID-19, since SARS-CoV-2 may be transmitted also in the pre-symptomatic period [39,59]. Moreover, the molecular and immunological diagnostic tools are useful for monitoring the efficacy of preventive and/or therapeutic tools. Finally, the viral genome sequencing capacity, which is rapidly improving, makes the early detection of variants possible. The progress represented by the next generation sequencing (NGS) multiplies the diagnostic possibilities, by simultaneously sparing time and costs. NGS makes an early identification of SARS-CoV-2 variants possible and may drive the research towards the identification of suitable antigens for diagnosis, vaccines and monoclonal antibodies (https://www.gao.gov/assets/720/713540.pdf, accessed on 10 April 2023).

### 5.2. Antibiotics

The production of antibiotics by bacteria was suggested by Louis Pasteur in the second half of XIX century and the first anti-infective chemotherapeutic agent, the anti-*Treponema pallidum* salvarsan, was developed at the end of XIX century by Paul Ehrlich. Sulfonamide drugs were made available in the thirties of the 20th century and the penicillin was discovered by Alexander Fleming in 1928, but it was only available as medicine starting with World War II [63].

During the “Spanish flu”, severe pneumonia, which represented the most frequent flu complication due to bacterial super-infection and the most frequent cause of death, was well known and recognized by the medical doctors. It can be speculated that, had antibiotics been available, they would have saved many lives, thus consistently reducing the mortality rates. Although the lung damaging in COVID-19 is mainly dependent on virus-induced local hyper-inflammation and not on bacterial super-infection, which is quoted to be around 7% [64], the availability of antibiotics represents a further therapeutic resource in such a minority of complicated COVID-19 cases, and another important difference between the two pandemics.

### 5.3. Antivirals

The history of antivirals started in the 1960s of the last century, more recently than that of antibiotics. Currently, more than one hundred drugs are approved by the Food and Drug Administration (FDA) for the treatment of human immunodeficiency virus (HIV), hepatitis C virus (HCV), hepatitis B virus (HBV), respiratory syncytial virus (RSV), herpesviruses, influenza virus, human papilloma virus (HPV) and SARS-CoV-2 [65,66].

In particular, remdesivir, an inhibitor of the viral RNA-dependent, RNA polymerase, which is intravenously administered, has been approved by FDA and European Medicines Agency (EMA) for adults and children with COVID-19 and pneumonia on the basis of demonstration of a faster recovery than placebo [67].

FDA and EMA were even granted emergency use authorization (EUA) to the association nirmatrelvir/ritonavir, as inhibitor of viral protease Mpro, which is required for viral replication, and molnupiravir, a nucleoside analogue inhibiting SARS-CoV-2 replication by viral mutagenesis. Both drugs are administered by oral route to non-hospitalized patients, and a real-life retrospective study has shown a very low rate of hospitalization, death, and adverse events to the drugs [68]. Recently it has been calculated that, had the nirmatrelvir/ritonavir been available in January 2022 during the surge of the Omicron variant, 4800 deaths would have been averted in the USA [69]. Conversely, the EMA approval of molnupiravir has currently been retired, on the basis of a negative evaluation by the Committee for Medical Products for Human Use (CHMP), pending a request of re-examination (https://www.ema.europa.eu/en/medicines/human/summaries-opinion/lagevrio, accessed on 3 June 2023).

However, the antivirals represent an important therapeutic tool, especially for frail, even non-hospitalized patients. Moreover, their mechanism of action does not involve the SP, and their activity against the VOCs, in contrast to the antiviral monoclonal antibodies (MoAbs), may not be reduced.

### 5.4. MoAbs and Biologic Molecules

MoAbs specific for pathogen’s antigens have recently been introduced for passive immunotherapy of infectious diseases for which an effective vaccine is not yet available, as RSV, or if the vaccine is poorly used and/or scarcely protective, as anthrax, or finally in addition to the vaccine, but with a different indication [70,71].

In particular, many MoAbs were developed against SARS-CoV-2, and some have already received the EUA by the FDA and/or EMA. Four MoAbs have been authorized by the EMA; they target the SP of SARS-CoV-2, thus preventing the virus from binding the ACE2 receptor to enter the body’s cells. These are tixagevimab/cilgavimab, casirivimab/imdevimab, sotrovimab, and regdanvimab. The first three MoAbs have even been authorized by the FDA, whereas, in place of regdanvimab, bebtelovimab was first authorized then revoked by the FDA. The EMA has issued advice on the use of a fifth product bamlanivimab/etesevimab) under Article 5 (3) of Regulation 726/2004. These MoAbs are indicated for treating frail patients, who do not require supplemental oxygen and some of them are also approved for the prevention of COVID-19 in exposed individuals; however, the EMA (and FDA) warn that MoAbs may not be effective against Omicron variant and subvariants (https://www.ema.europa.eu/en/news/etf-warns-monoclonal-antibodies-may-not-be-effective-against-emerging-strains-sars-cov-2#:~:text=EMA%20has%20issued%20advice%20on,do%20not%20require%20supplemental%20oxygen, accessed on 10 April 2023). A Cochrane analysis on the protective effect of tixagevimab/cilgavimab and casirivimab/imdevimab against the VOCs excepting Omicron, in pre-exposure and post-exposure trials, showed a reduction of infections and hospitalizations; however, it could not show definitive evidence of protection [72]. The emergence of the Omicron variant and the several subvariants, characterized by many mutations at the SP level, has profoundly modified the antigenic protein structure, thus markedly reducing the recognition by and the effectiveness of MoAbs, and only sotrovimab maintains a protective activity [73].

The emergence of variants with mutations in the epitopes recognized by MoAb represents a weakness for their general use and a point in favor of antivirals that with higher probability maintain their effectiveness against the variants and subvariants (https://www.ema.europa.eu/en/documents/public-statement/etf-statement-loss-activity-anti-spike-protein-monoclonal-antibodies-due-emerging-sars-cov-2_en.pdf, accessed on 10 April 2023).

Other biological molecules, such as tocilizumab, a humanized MoAb directed against the interleukin (IL)-6 receptor, and anakinra, a biological human receptor antagonist of the IL-1 receptor, have been repurposed by the EMA for COVID-19 patients, even though they are not pathogen specific. In particular, “tocilizumab can also be used in adults with COVID-19 who are receiving treatment with corticosteroid medicines by mouth or injection and require extra oxygen or mechanical ventilation (breathing assisted by a machine)” [(https://www.ema.europa.eu/en/medicines/human/EPAR/roactemra) accessed on 10 April 2023]. Anakinra “can also be used in COVID-19 adults with pneumonia requiring supplemental oxygen (low or high flow oxygen) and who are at risk for developing severe respiratory failure, as determined by blood levels of a protein called suPAR (soluble urokinase plasminogen activator receptor) of at least 6 ng per ml” (https://www.ema.europa.eu/en/medicines/human/EPAR/kineret, accessed on 10 April 2023). These supporting treatments, together with dexamethasone, represent further therapeutic resources in severe COVID-19 infections, which were lacking over one century ago.

### 5.5. Vaccines

“Spanish flu” more than one century ago could not benefit from a specific vaccine against the influenza virus. The virus of influenza was still unknown, it was only discovered in 1933 and its isolation put the basis for the production of a live influenza vaccine in 1936 [74,75] and an inactivated vaccine in 1940 [76]. A mixed bacterial vaccine to prevent the secondary bacterial pneumonia, which was generally responsible for death, was used with apparent success [32]. Several trials were performed to check the effectiveness of these bacterial vaccines, but their results were generally inconsistent and sometimes contradictory [77], although it cannot be excluded that bacterial vaccines may have prevented part of cases of influenza-associated pneumonia [78].

Conversely, COVID-19 could benefit from an unprecedented economic and research effort, which allowed to develop and make available, in the very short time of less than one-year, effective and innovative vaccines, which have markedly reduced the severity and mortality of the pandemic [79,80,81]. It should be underlined that the use of a specific vaccine to contain a pandemic from a respiratory virus is an absolute novelty, considering that all the flu pandemics, including the last one in 2009, could not benefit from the use of specific vaccines [1], because the traditional vaccines, unlike the innovative mRNA and viral vectored vaccines, cannot be prepared in short time, thus they could not timely be prepared in order to be ready for use at the beginning of the pandemic. COVID-19 vaccines have proven to be effective, particularly in reducing hospitalization, intensive care unit, and death, even in frail patients [82]. Unfortunately, COVID-19 vaccines are less effective than expected in preventing infection, especially when the VOCs, in particular Omicron, have emerged [73,83,84], but this limitation cannot be considered a vaccine failure [84]. Risk reduction of hospital admission and severe disease induced by immunological protection conferred by both, disease and vaccination, is stronger and more persistent than protection against reinfections, which is lower and wanes rapidly, within months [85,86]. In fact, antibody protection was shown to be short-lived, making booster doses required to resume acceptable levels of circulating neutralizing antibodies (nAbs) [87]. However, the developed vaccines, mostly based on the mRNA or adenoviral vector technology, may induce protection by promoting the expansion of cellular in addition to humoral immunity. The contribution of humoral immunity to the vaccine efficacy is dependent on the expansion of B lymphocytes and plasma cells secreting nAbs. These nAbs, recognizing the epitopes on the viral glycoproteins required by the viruses to enter and infect the target cells, i.e., the RBD of SP for SARS-CoV-2, inhibit the infection of target cells. The mutations occurring in the RBD of new VOCs of SARS-CoV-2 may limit the efficacy of nAbs generated by the vaccine based on the sequence of the original Wuhan virus [88]. In addition, since SARS-CoV-2 is a virus that may infect macrophages [89], vaccines produced to protect from COVID-19 should not favor viral opsonization by non-nAbs, that, in turn, would promote the infection of macrophages, ultimately leading to an antibody dependent enhancement (ADE) of infection phenomenon [90,91]. The contribution of cellular immunity to the vaccine efficacy is mainly mediated by cluster of differentiation (CD)4+ and CD8+ T lymphocytes. Antigen-specific CD4+ are helper T cells required to permit the expansion of specific B and CD8 + T lymphocytes and are generally expanded by all the vaccines. Antigen-specific CD8+ lymphocytes are a pleiotropic population of cells that includes cytotoxic cells able to kill virus infected cells. Due to the specific mechanisms that govern CD8+ T cell expansion, i.e., the generation of immunogenic peptides associated to major histocompatibility complex (MHC) class I molecules in antigen presenting cells [92], not all the vaccines are able to cause the expansion of killer T cells. This capacity is limited to live attenuated viral vaccines or genetic vaccines based on DNA or mRNA, and generally it is not shared by subunit vaccines [93]. Thus, even with the most recent variants, if antibody protection from infection may be considered less efficient, the same do not seem to occur at level of cellular immunity. T cells recognize short, 8– to 15–amino acid linear peptides, derived from the proteasomic degradation of nascent viral proteins, that are not limited to the RBD domain of SP, where most mutations occur. Therefore, T cell responses remain largely intact against most variants with >80% of T cell epitopes conserved [94,95]. Moreover, if escape from a T cell epitope occurs, differences in human leukocyte antigen (HLA)-peptide presentation suggest that a mutation that causes escape from T cell immunity in one person is unlikely to do so in another person. Overall, emerging viral variants substantially affect antibody neutralization, but may have a minimal impact on T cell responses [96,97]. This phenomenon is likely to be the reason why a protection against severe disease and death is observed in vaccinated individuals that are infected with the Omicron variant compared with non-vaccinated [97].

Although the vaccines have shown a good efficacy against disease severity and death, the reduced capacity to protect from infection, the short half-life of nAbs, as well as the emergence of VOCs, and in particular, the Omicron variant, make the contribution of the vaccine to achieve pandemic containment less efficient than foreseen. Furthermore, it was observed that a booster dose of the recent bivalent vaccine, containing the original Wuhan viral strain and the Omicron BA.4/BA.5 sequences, fails to induce a significant increase of immune response to the Omicron variant compared with the previous vaccine containing the original Wuhan strain sequence only. A possible explanation is suggested by the phenomenon of the “original antigenic sin” [98], a theory proposed by Thomas Francis Jr in 1953 [99], after observing the serum antibody titer of normal subjects against different influenza viral strains in the period 1943–1951. The results showed that the viral strain encountered in infancy primed the immune response and such immune imprinting remained prevalent throughout life following stimulation by other partially cross-reactive influenza viral strains. Thus, although the immune response to the other partially cross-reacting viral strains was modest, it was able to powerfully boost the immune response to the priming viral strain, so maintaining a high level of immune response to the priming antigen for life, without a robust and lasting protective response against the new antigenic variants. This phenomenon has been repeatedly observed by different authors with influenza [100,101], other viruses [102], and more recently, even in the neutralizing antibody response to SARS-CoV-2 Omicron variant bivalent vaccine, which was not significantly higher than the one induced by the monovalent vaccine composed by the original Wuhan strain, associated with a lack of increase of T-cell response [98,103,104,105,106,107,108]. A phenomenon similar to the “original antigenic sin” has even been described for cytotoxic T lymphocytes [109]. Possible strategies to overcome the immune imprinting consist in the use of adjuvants [110] or antigenically distant molecules compared with the imprinting antigen [111]. However, the high antigenic variability makes the response to vaccine less effective, particularly in consideration of the need that these vaccines should periodically be adapted to the circulating viral strains and repeated, thus chasing and not anticipating the viruses [98]. In addition, for COVID-19, the possible immune imprinting by the other α- and ß-coronaviruses, widely circulating and responsible for the common cold, adds matter to the “original antigenic sin”, which is not uniquely applied in the context of the variants compared to the original Wuhan viral strain, but even to coronaviruses other than SARS-CoV-2 [112]. Finally, for COVID-19, the possible ADE that is an antibody-dependent help to virus for infecting host cells has been hypothesized, as observed for other viral infections or vaccines, such as dengue virus and vaccine, RSV and measles inactivated vaccines, feline, simian, and human immunodeficiency virus vaccines, SARS-CoV and MERS-CoV vaccines [113]. However, ADE has not been observed during either COVID-19 infection or vaccination [114,115].

At least in western countries, even if an extraordinary vaccination campaign led to an unprecedented rapid mass immunization, vaccine hesitancy or even opposition to vaccinations were observed; in case of the new COVID-19 vaccines, uncertainty of long-term side effects [116] may have contributed to partially limit the global immunization program. Thus, for all the above reported considerations, although COVID-19 vaccines have been very useful for reducing case-fatality rate and the pandemic-associated congestion, with the possible collapse of the hospitals, they hardly may contribute to contain the pandemic. However, the innovative COVID-19 vaccines have shown to represent a real step forward in the history of vaccination, because, for the first time in the history of pandemics, their relatively easy preparation and modification have allowed a pandemic to be promptly faced by a specific vaccine [1]. Moreover, recently a tentative universal mRNA flu vaccine has been developed, which has demonstrated to be highly promising at the experimental level [117,118]. In fact, a possible resolution of reduced vaccine effectiveness as a consequence of viral antigenic variability may only come from the development of universal vaccines for influenza [117,118,119] and COVID-19 [120], able to confer protection from infection. Unfortunately, this is not an easy target to reach in consideration of the great biological and technical problems. However, universal vaccines may represent a useful tool for pandemic containment [109], but cannot be developed to eradicate zoonotic diseases, such as influenza and COVID-19. The eradication of a disease by vaccines occurred for smallpox, which was declared extinguished in 1980, and is expected for polio and measles, which represent the next objectives of eradication by the WHO, but these are viral diseases with no known animal reservoir.

It has to be underlined that Omicron shows an increased contagiousness, which was responsible for the three pandemic waves in 2022, counterbalanced by a milder clinical course and a reduced mortality, which may only be observed in frail patients, thus allowing us to foresee that even for COVID-19, similarly to influenza, the natural evolution may be predicted towards an endemic, generally mild, respiratory disease. The milder clinical course of the Omicron variant, which presents a high antigenic distance from the other VOCs and from the ancestral original viral strain [121], is probably due to intrinsic characteristics, but even to the interaction with vaccinated and/or previously infected, thus partially protected, hosts [97]. This is a further similarity between the two pandemics, which suddenly started with a very severe and aggressive clinical course, burdened by a high mortality, to evolve towards a different, low-severity and low-mortality, endemic disease. Although herd immunity may hardly be achieved in respiratory diseases caused by highly variable viruses, such as influenza and COVID-19 [122], the relative rapid kinetics of transition from the pandemic to the endemic state in “Spanish flu” allows one to hypothesize that this might have been due to the progressive increase of immunity in survivors. There is evidence, indeed, that immunity induced by the spring wave of the 1918 pandemic influenza did not wane rapidly, and was protective against the deadly second wave in October. As a matter of fact, most of the American soldiers who contracted the mild form of influenza in the spring escaped the fall wave; conversely, military units with no history of exposure to the spring wave reported a very high attack rate of influenza during the fall outbreak. Moreover, some soldiers who contracted influenza both in spring and in fall showed, as a rule, a mild disease during the second wave [123]. The infection-induced immunity during the “Spanish flu” was probably more protective than the immunity induced by COVID-19 vaccines, which reduce but cannot interrupt viral circulation. It must be underlined that the higher protection observed during the “Spanish flu” may be a consequence of a higher antigenic stability of the influenza virus, which prevented any immune escape. Conversely, the high variability of SARS-CoV-2, which has generated many escaping variants and subvariants, a phenomenon never observed before during the influenza pandemics, may probably be due not only to an intrinsic characteristic of the virus, but even to the host immune pressure [124].

## 6. Long-Term Sequelae

Another aspect to be discussed regarding the two pandemics is the observation of the long-term sequelae of the acute infection. In fact, for SARS-CoV-2 infection the “long COVID”, a multisystemic condition following an acute COVID-19 infection, probably due to SARS-CoV-2 persistence in the host and reactivation by unrelated viruses or other stimuli [125], has been observed in 10–30% of non-hospitalized, 50–70% of hospitalized, and 10–12% of vaccinated cases [126]. In this multisystemic condition, fatigue is a very frequent, nearly constant symptom, thus evoking the chronic fatigue syndrome (CFS), which may be observed as sequelae of different viral diseases [126]. For “Spanish flu”, it is difficult to find evidence documenting a clinical situation reminding the long COVID, even in consideration of the social and environmental disruption following the ongoing World War I; however, a post-influenza fatigue resembling what is today defined as CFS was observed following the ”Spanish flu” [127]. Moreover, retrospective studies showed an increase in cardiovascular diseases in cohort subjects born in 1919, thus suggesting that the *in utero* or post-natal exposure to “Spanish flu” virus could have determined long-term cardio-vascular sequelae [128] (Table 2).

## 7. Conclusions

Although more than one century has passed from the “Spanish flu” to the onset of the COVID-19 pandemic, the ignorance on the origin of the viruses and the difficulties in understanding the reasons for the high severity and mortality of both pandemics are the same. A similarity can also be observed in the natural history of the two pandemics, which showed the tendency to evolve towards endemic milder diseases, with kinetics which appears similar, but slower in COVID-19, despite the availability of vaccines, MoAbs and antivirals.

Although unproven and impossible to demonstrate during the “Spanish flu”, the survivors could have developed a protective immunity, which worked as herd immunity, that contributed to the end of the pandemic after three waves, even if at extremely high cost in terms of human lives. It is only possible to hypothesize that, had antibiotics been available in 1918, the majority of secondary pneumonia-induced deaths would have been avoided. Conversely, in the COVID-19 pandemic, neither the disease nor the vaccination showed the capacity to induce herd immunity. The protective effect of the vaccines against severe diseases and mortality allowed to markedly reduce the congestion of the hospitals and the number of deaths, but the influence on the pandemic containment, thus on the transition from pandemic to endemic condition, appeared to be scarcely or no relevant.

The dynamic interaction between humankind and microorganisms is witnessed by the continuous description of new pathogens for humans [129,130]; influenza and COVID-19 pandemics are zoonotic diseases which originate from very close contact between humans and animals in every-day life in some countries. Only the adoption of more careful and responsible behaviors will facilitate the prevention of these terrible diseases.

## Figures and Tables

**Table 1 pathogens-12-00868-t001:** Biological characteristics of influenza viruses and coronaviruses.

Characteristics	Influenza Viruses	Coronaviruses
Nucleic acid	(−) ss-RNA	(+) ss-RNA
Family	*Orthomyxoviridae*	*Coronaviridae*
Types/genera	A-B-C-D	α-ß-γ-δ
Vaccine antigens	Hemagglutinin (HA, H) Neuraminidase (NA, N)	Spike protein (SP)
Pathogen for humans	A-B-C Pandemic virus A subtypes H1N1, H2N2, H3N2, Sporadically, avian viruses H5N1, H7N7, H7N9, H9N2	α (299E-NL63)-ß (HKU1-OC43, SARS-CoV, MERS-CoV, SARS-CoV-2)
Genome	Segmented (8 [A,B] and 7 [C,D] gene segments)	Non-segmented
Variability mechanisms	Reassortment (shift), mutations (drift)	Recombination, mutations
Genome mutation rate	Human flu A virus: 5.57 × 10^−3^ substitutions/site/year	SARS-CoV-2: 1.12 × 10^−3^ mutations/site/year
Variants	Theoretically 198 (18 HA × 11 NA), observed 120	Alpha-Beta-Gamma-Delta-Omicron
Animal reservoir	Waterfowl birds; mammals, including humans, swine	Bat

**Table 2 pathogens-12-00868-t002:** Similarities and differences between “Spanish flu” and COVID-19 pandemics.

Characteristic	Spanish Flu	COVID-19
Type of disease	Respiratory	Respiratory
Virus	*Orthomyxoviridae*	*Coronaviridae*
Transmissibility R_0_	2	2.5
Incubation period	1–4 days	2–14 days
Mortality	50–100 million	6,881,955 (10 March 2023)
Global average case-fatality rate	>2.5%	1.01%
Higher mortality in population	<65-year-old	>65-year-old
Cause of death	Secondary bacterial pneumonia	Hyperactive immune system
Place of origin	Kansas (USA)	Wuhan (China)
Long-term sequelae	Cardiovascular diseases/CFS	Long COVID
Vaccines	Anti-bacterial	Anti-SARS-CoV-2
Antibiotics	No	Yes
Antivirals	No	Yes
Passive immunotherapy	Plasma from convalescents	Convalescent plasma-MoAbs

CFS = chronic fatigue syndrome; MoAbs = monoclonal antibodies.

## Data Availability

Not applicable.

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
