# Peer review of "Have Diagnostics, Therapies, and Vaccines Made the Difference in the Pandemic Evolution of COVID-19 in Comparison with “Spanish Flu”?"

_pathogens, 2023, doi:10.3390/pathogens12070868_

Round 1
Reviewer 1 Report (Previous Reviewer 3)
. There are some misuses of the word "however" throughout the text, even when the sense is different from what this word represents. Please, check carefully for these examples as we can observe on the lines 30, 177, etc.
. Line 30: Change “However, most comparison studies…” to “Most comparison studies…”
. Line 59: Change “family of Coronaviridae” to “family Coronaviridae”.
. Line 81: Change “which is characteristic molecule” to “which is a characteristic molecule”
. . Line 177: Change “However, it has recently been found…” to “It has recently been found…”
. Line 177-180: “However, it has recently been found that a substitution in a single…viral strain to humans” - This sentence seems to be completely out of context in the text.
. As recently stated by the WHO, the authors should mention during the text that COVID-19 no longer constitutes a public health emergency of international concern.
-------------------------------------------------------------------------------------
Apart from these minor problems above, the quality of the paper has been overwhelmingly increased
Author Response
Reviewer 1
There are some misuses of the word "however" throughout the text, even when the sense is different from what this word represents. Please, check carefully for these examples as we can observe on the lines 30, 177, etc.
Line 30: Change “However, most comparison studies…” to “Most comparison studies…”
Line 59: Change “family of Coronaviridae” to “family Coronaviridae”.
Line 81: Change “which is characteristic molecule” to “which is a characteristic molecule”
Line 177: Change “However, it has recently been found…” to “It has recently been found…”
Line 177-180: “However, it has recently been found that a substitution in a single…viral strain to humans” - This sentence seems to be completely out of context in the text.
As recently stated by the WHO, the authors should mention during the text that COVID-19 no longer constitutes a public health emergency of international concern.
Apart from these minor problems above, the quality of the paper has been overwhelmingly increased
Answer to Reviewer 1
We thank Reviewer 1 for her/his comment and modified the text as suggested.
Regarding the remarks raised at line 177 and the phrase at lines 177-180, which seems completely out of context in the text to the reviewer, we changed the text in order to make it easier to explain what we meant and justify the the opportunity to leave the word “however”.
Reviewer 2 Report (Previous Reviewer 2)
Lista et al. submitted the corrected version of the manuscript, which should compare the pandemic of Spanish flu and COVID-19. What is important, authors do not rely on the scientific data, but 'speculate', as they wrote in the Introduction (Line 53-55). I can imagine that it is extremely hard to compare pandemic, which happened in completely different era, with completely different antiviral strategies, antiviral drugs and vaccines. Unfortunately, I am still not satisfied, the paper is chaotic and show some differences between two pandemics, without any conclusions.
Author Response
Reviewer 2
Lista et al. submitted the corrected version of the manuscript, which should compare the pandemic of Spanish flu and COVID-19. What is important, authors do not rely on the scientific data, but 'speculate', as they wrote in the Introduction (Line 53-55). I can imagine that it is extremely hard to compare pandemic, which happened in completely different era, with completely different antiviral strategies, antiviral drugs and vaccines. Unfortunately, I am still not satisfied, the paper is chaotic and show some differences between two pandemics, without any conclusions.
Answer to Reviewer 2
We read with attention the comments of Reviewer 2 and we agree that it is extremely hard to compare pandemic, but we felt it was worthwhile to speculate on the differences and similarities to learn lessons useful for possible new emergences. We do not agree with Reviewer 2 in describing as chaotic our paper and when she/he writes that we do not provide conclusions. The paper is organized in different areas of discussion and the conclusions are speculative, since it is impossible to provide direct proofs.
Reviewer 3 Report (New Reviewer)
This is important article that compares two major pandemics in the past 100 years. I believe this article provides a good overview of influenza and SARS-COV-2 viruses.
Here are minor suggestions I have for the improvement:
Introduction
Please use the language such as “this study aims”, or “the objective of this study is” instead of “we decided”.
Viruses
“Two different single-stranded (ss)-RNA viruses caused the pandemics: influenza vi- 57 rus, that pertain to the family Orthomyxoviridae [10], and Severe Acute Respiratory Syn- 58 drome CoronaVirus (SARS-CoV)-2, classified in the family of Coronaviridae [9].”
Please specify which of the mentioned viruses caused which pandemic, because this sentence is confusing without listing the specific pandemic each of the mentioned viruses caused.
5.2. Antibiotics
“Although the lung damaging in COVID-19 is mainly dependent on virus induced local hyper-inflammation and not on bacterial super-infection, the availability of antibiotics represents a further therapeutic resource in some complicated COVID-19 case…”
Please provide a citation indicating that antibiotics represent a therapeutic resource in COVID-19 cases.
Author Response
Reviewer 3
This is important article that compares two major pandemics in the past 100 years. I believe this article provides a good overview of influenza and SARS-COV-2 viruses.
Here are minor suggestions I have for the improvement:
Introduction
Please use the language such as “this study aims”, or “the objective of this study is” instead of “we decided”.
Viruses
“Two different single-stranded (ss)-RNA viruses caused the pandemics: influenza virus, that pertain to the family Orthomyxoviridae [10], and Severe Acute Respiratory Syndrome CoronaVirus (SARS-CoV)-2, classified in the family of Coronaviridae [9].”
Please specify which of the mentioned viruses caused which pandemic, because this sentence is confusing without listing the specific pandemic each of the mentioned viruses caused.
5.2. Antibiotics
“Although the lung damaging in COVID-19 is mainly dependent on virus induced local hyper-inflammation and not on bacterial super-infection, the availability of antibiotics represents a further therapeutic resource in some complicated COVID-19 case…”
Please provide a citation indicating that antibiotics represent a therapeutic resource in COVID-19 cases.
Answer to Reviewer 3
We thank Reviewer 3 for her/his comment and modified the text as suggested.
Reviewer 4 Report (New Reviewer)
This is an excellent review. The authors not only comprehensively list the similarities but also the differences between these two pandemics in terms of the viruses themselves, the use of diagnostics, and the antivirals for therapeutics, and the development and supplies of vaccines for disease prevention and deployment of public health containment measures. The authors also judiciously use tables to summarise the highlights, thus making it easier for the readers to follow. Last but not least, the authors offer their expert opinions on a number of important aspects regarding the two pandemics, with which this reviewer happily concurs.
Author Response
Reviewer 4
This is an excellent review. The authors not only comprehensively list the similarities but also the differences between these two pandemics in terms of the viruses themselves, the use of diagnostics, and the antivirals for therapeutics, and the development and supplies of vaccines for disease prevention and deployment of public health containment measures. The authors also judiciously use tables to summarise the highlights, thus making it easier for the readers to follow. Last but not least, the authors offer their expert opinions on a number of important aspects regarding the two pandemics, with which this reviewer happily concurs.
Answer to Reviewer 4
We thank Reviewer 4 for her/his comment.
Round 2
Reviewer 2 Report (Previous Reviewer 2)
Accept in present form
This manuscript is a resubmission of an earlier submission. The following is a list of the peer review reports and author responses from that submission.
Round 1
Reviewer 1 Report
The authors compare the Spanish flu and COVID-19 pandemic, focusing therapies, diagnostics and vaccines. It is hard to calculate the precise number and compare deaths after the flu pandemic and COVID-19 pandemic one century later. Nevertheless, this comparison confirms the importance of vaccines.
However, the benefits of vaccines still being debated. Therefore, studies analyzing vaccines and the health benefits of vaccination are important not only for the research community, but for the society.
To increase the significance of this work, the authors should highlight the main goal of vaccination: The protection against death. Even when vaccinated people get sick, this not means that the vaccine was not successful. The main goals of vaccines are to avoid hospitalization and death.
The COVID-pandemic was an example, how crucial is the communication between politicians and the population, to convince as many people as possible, how important are vaccines. This deeper discussion would impact in this review.
Reviewer 2 Report
The ongoing COVID-19 pandemic is quite often compared with the biggest XX century pandemic of Spanish flu, even though that through the time between those pandemic, the epidemiological, diagnostic and treatment standards have significantly changed. Lista et al. in their manuscript tried to compare the diagnostic, therapies and vaccines between the Spanish flu and COVID-19 pandemic. Unfortunately, I did not find any comparison of the diagnostic nor therapies for the above mentioned pandemic. There is quite large section on the vaccines, however it should be highlighted that the influenza vaccines was firstly developed in 1930, so it is meaningless to compare this aspect. In addition, the whole manuscript is quite chaotic and hard to follow.
COMMENTS
1. The authors contradicted themselves in the abstract. It is written that ‘Comparisons have been carried out in the first period of COVID-19, when the vaccines were either not yet available or had been only used for a short period’ (Page 1 Lines 28-29), and in the next sentence you can find that they ‘wanted to analyze the role that (…) innovative COVID-19 vaccines (…) may have had in the pandemic containment.’ So, what was the aim of the manuscript?
2. The SARS-CoV-2 is not ‘highly variable’ virus (Page 1 Line 27)
3. ‘The natural reservoir for influenza virus is represented by the waterfowl birds (excepting HA17NA10 and HA18NA11, which have only been observed in Peruvian bats); however, even mammals, including humans and swine, may be infected.’ (Page 2 Lines 63-65) – needs citation
4. ‘Among 65 the theoretical 198 possible subtypes, only 120 have been actually found in nature, and 66 only three, namely H1N1, H2N2 and H3N2 have been observed in humans’ (Page 2 Lines 65-67) – how about human infection with avian H5N1, H7N1, etc.?
5. ‘Conversely, coronaviruses are positive-sense ss-RNA viruses with a compact, unseg-74 mented genome, which enter the host cells via the angiotensin-converting enzyme (ACE)2 receptor recognition by the receptor binding domain (RBD) area, located in the tract S1 of the Spike protein, which is characteristic of coronaviruses.’ – incorrect statement, not all coronaviruses entry is mediated via ACE2, i.e. the MERS-CoV host cell receptor is DPP4
6. ‘The rate of variability is quite high for both viruses, as in general for RNA viruses, whose RNA polymerase has a very lower capacity to check for possible replication mistakes than the DNA polymerase, which is present in the DNA viruses, that are, therefore, more stable’ (Page 2 Lines 90-92) – please be more specific and indicate the genome mutation rates for both viruses. In addition, it should be highlighted that the SARS-CoV-2 RNA polymerase has proofreading exonuclease activity.
7. ‘Additional flu pandemics occurred, the first, A(H1N1) Russian flu, in 1977, and the second, A(H1N1)pdm09 pandemic flu, more recently in 2009.’ (Page 2 Lines 104-106) – unclear sentence!
8. Table 1 – ‘genome – Influenza viruses – segmented (8 (…) genes)’ – incorrect statement! There are 8 genome segments in IAV, and 8th segment encode at least two genes, i.e., NS1 and NS2, so the IAV needs to include more than 8 genes.
9. ‘Although reliable 149 data are not available, it is estimated that “Spanish flu” infected approximately 500 million subjects, one third of the world’s population in 1918, with a mortality of approximately 50 million cases’ (Page 4 Lines 149-151) according to this statement and some straight-forward calculation, the fatality rate is 10%. Few lines above you mentioned that ‘its death rates were 2-125 4%’ (Page 3 Lines 125-126). Please explain these discrepancies.
10. ‘The highest wave was the first one of 173 2022, whereas the highest mortality has been observed in the second half of 2020, just before the start of vaccine administration. These data clearly indicate how vaccines have shown to be effective against mortality, whereas they are scarcely effective in preventing infections, particularly by the Omicron variant, thus being unable to interrupt the virus circulation.’ (Page 4 Lines 173-177) – incorrect statement. You cannot estimate the vaccines efficiency based on the mortality data before the vaccine development. The following pandemic waves were weaker not only due to the vaccination, but also due to the natural acquired immunity in the population and emergence of omicron variant.
11. ‘However, it may be expected that these measures could have been more effective in the 184 case of “Spanish flu” compared with COVID-19, for the possible transmissibility of SARS-185 CoV-2 during the incubation period [29], which reduces the effectiveness of these 186 measures.’ - do you really believe that early XX century protective masks, hand hygiene, isolation and quarantine were more effective against infection than during the COVID-19 pandemic in XXI century?
12. I do not see any sense in the whole section ‘Vaccines’. How do the authors compare the time when the anti-influenza vaccine was not even developed with the times when the mRNA in the liposome vaccine was used? In addition, this section is very chaotic and hard to follow. The authors jumped from one topic to another without any logic.
Reviewer 3 Report
. Line 27 - "These pandemics share similarities, being both caused by highly variable and transmissible respiratory RNA viruses, and diversity." - diversity of what?
. Sometimes the construction of some sentences goes very much confusing (although not gramatically incorrect), taking a lot of time to the reader comprehend what the authors really meant. Most of these times are a result of extremely long sentences in which the authors do not establish a focus in the information they want to pass on. Some examples are:
- Although early diagnosis, therapies, anti-COVID-19 vaccines have markedly reduced the pandemic severity and mortality, thus preventing the collapse of the public health services, their influence on the reduction of infections and re-infections, thus on the transition from pandemic to endemic condition, appears to be of minor relevance.
- Influenza is a negative-sense ss-RNA virus and its genome is segmented in 8 different genes in types A and B, 7 in types C and D, 2 of which, in the type A, code for either 18 different hemagglutinins (HA or H) or 11 different neuraminidases (NA or N), the two glycoproteins needed for attachment to and detachment from the host cell, respectively.
- Shifts are deep viral genetic modifications, which are generally associated with pandemics, that may occur with a variable periodicity, and may be complete, involving all the gene segments, as in the subtype H1N1“Spanish” pandemic, or limited to some gene segment, such as in the case of Asian subtype H2N2 pandemic in 1957 , that exhibited only three new genetic segments and the Hong Kong subtype H3N2 pandemic in 1968 exhibited only two new genetic segments).
. Line 65-66: "Among the theoretical 198 possible subtypes, only 120 have been actually found in nature" - I wouldn't use the word "only" in this case.
-------------------------------------------------------------------------------------
Although presenting some interesting considerations related to both pandemics, the authors have strongly failed to show what was suposed to be their main objetive. As they themselves describe in the text:
"We wanted to analyze the role that the advanced diagnostics, innovative COVID-19 vaccines, and anti-viral agents, including monoclonal antibodies, may have had in the pandemic containment."
From my point of view nothing was analyzed throughout the text. No data was shown comparing both epidemics. What they really have done in this work was to brought information (some of which not even relevant to what they have proposed) concerning both of these epidemics and comparing some of these aspects using generic sentences.
I really missed what they brought in their title "Have diagnostics, therapies, and vaccines made the difference in COVID-19 compared with “Spanish flu” pandemic?".
Just as an example, the part mentioning diagnostics do not even appear on their text. That being said, as I couldn't see their objetives fully present on the text, I can't recommend this work for publication.
-------------------------------------------------------------------------------------